# Perfluorooctanoic Acid and Perfluorooctane Sulfonate in Human Milk: First Survey from Lebanon

**DOI:** 10.3390/ijerph20010821

**Published:** 2023-01-01

**Authors:** Hussein F. Hassan, Haneen Bou Ghanem, Joelle Abi Kharma, Mohamad G. Abiad, Jomana Elaridi, Maya Bassil

**Affiliations:** 1Nutrition Program, Natural Sciences Department, Lebanese American University, Beirut 1102-2801, Lebanon; 2Department of Nutrition and Food Sciences, Faculty of Agricultural and Food Sciences, American University of Beirut, Beirut 1107-2020, Lebanon; 3LEAF—The Laboratories for the Environment, Agriculture and Food, American University of Beirut, Beirut 1107-2020, Lebanon; 4Chemistry Program, Natural Sciences Department, Lebanese American University, Beirut 1102-2801, Lebanon; 5Human Nutrition Department, College of Health Sciences, QU Health, Qatar University, Doha 2713, Qatar

**Keywords:** PFOS, PFOA, human milk, diet

## Abstract

Human milk is the primary source of nutrition for infants in their first year of life. Its potential contamination with perfluorooctanoic acid (PFOA) and perfluorooctane sulfonate (PFOS), a group of toxic man-made chemicals, is a health concern that may threatens infants’ health. Our study aims to assess the levels of PFOA and PFOS in the breast milk of Lebanese lactating mothers and the maternal factors associated with their presence. High-performance liquid chromatography (HPLC) coupled with a Micromass Quattro micro API triple quadrupole mass spectrometer was used to detect the level of contamination in 57 collected human milk samples. PFOA and PFOS were present in 82.5% and 85.7% of the samples, respectively, while PFOA levels ranged between 120 and 247 pg/mL with a median of 147 pg/mL, and those of PFOS ranged between 12 and 86 pg/mL with a median of 27.5 pg/mL. The median contamination for PFOA exceeded the threshold set by the European Food Safety Authority (EFSA) (60 pg/mL); however, that of PFOS was below the threshold (73 pg/mL). The consumption of bread, pasta, meat, and chicken more than twice per week and that of white tubers and roots at least once per week was significantly associated with higher levels of PFOA (*p* < 0.05). No significant association was found between maternal age, BMI, parity, level of education, place of residence, source of water used, and smoking with the levels of PFOA and PFOS in the human milk. Additionally, the consumption of cereals at least twice per week was significantly associated with higher levels of PFOS. These findings call for actions to improve the local environmental and agricultural practices, and the regulations and standards for inspecting imported food. It is important to highlight that the benefits of breastfeeding outweigh the reported contamination with PFOS and PFOA in our study.

## 1. Introduction

Per- and polyfluoroalkyl substances (PFAS) are a complex group of anthropogenic chemicals containing more than 9000 distinct compounds [1,2]. These chemicals have been extensively used since the 1940s due to their unique hydrophobic and lipophobic chemical properties, making them extremely stable, and repellents for water, grease, and stains [3,4,5]. Due to these unique characteristics, PFAS are still under use in several countries, with serious consequences [3]. They are utilized to manufacture several products, including surfactants, lubricants, polishes, Teflon products, textiles, furniture, carpets, papers, and firefighting foams [1,6]. The effect of PFOA has been evaluated in an animal model-based study which evaluated the expression of several target genes in carps exposed to environmental doses of PFOA. Thus, even in conditions of low concentrations, the exposure to these compounds can negatively impact the expression of genes in animals and possibly humans [7].

Perfluorooctanoic acid (PFOA) and perfluorooctane sulfonate (PFOS) are the most studied compounds among all PFAS. Over the last few decades, they have received global attention due to their widespread use, high stability, and toxicity [8,9]. Several toxicological and epidemiological studies have shown that these toxic compounds act as endocrine disruptors and have detrimental health consequences, including nephrotoxicity, hepatotoxicity, immunotoxicity, in addition to developmental toxicity [10,11].

According to [12], it is estimated that the half-life for PFOS in humans is 5.4 years, and that of PFOA is 3.8 years. In addition to their long half-lives, these substances have a low elimination rate from the human body, posing a severe health threat [13]. Therefore, these substances are considered persistent and bio-accumulative environmental contaminants due to their high stability and resistance to various forms of degradation, including biological, chemical, and thermal, making them a public health concern [14,15,16].

Due to their extensive use and bioaccumulation properties, several studies have shown that PFOA and PFOS are now present in the serum of humans worldwide [4]. Human exposure to these stable contaminants occurs through various pathways. Using PFAS-containing products is the major route of exposure [17]. However, diet is also considered a significant source of exposure to PFAS, with fish and shellfish being the primary dietary sources. Meat products, eggs, dairy, and vegetables are other potential sources that can become contaminated due to soil, water, animal feed, or packaging contamination [18]. Tap water has also been found to contain a significant amount of PFAS [19]. In addition to their presence in water, they are also found in air, sediments, and wildlife, increasing the risk of human exposure through air inhalation, dust ingestion, and dermal contact [20].

In addition, it was reported that human exposure to PFAS begins during gestation and continues during breastfeeding, leading to severe health outcomes that are mainly associated with immunotoxicity, hepatotoxicity, neurobehavioral toxicity, reproductive toxicity, lung toxicity, and hormonal effects [1,21,22].

Human milk is the best source of nutrition for infants providing them with the optimum quality and quantity of nutrients to grow. In addition, it offers unique immunological benefits, decreasing their risk of acute illnesses and chronic diseases later in life [23]. According to the World Health Organization (WHO), exclusive breastfeeding is recommended for the first six months of life and should continue up to two years with proper complementary feeding [24]. However, human milk might be at risk of several environmental contaminations, including PFAS, due to maternal exposure before and during pregnancy and throughout her life [25,26]. It was reported that breastfeeding accounts for 94% of the exposure to PFAS by infants younger than six months of age [27]. Previous studies showed that the duration of breastfeeding is strongly correlated with the levels of PFAS in children, where breastfed children have two folds the concentration of these contaminants in their blood compared to non-breastfed ones [28].

It is still unclear how PFOA and PFOS are transferred from the mother’s blood to her breast milk. However, it has been shown that these perfluorinated substances are bound to proteins in maternal blood [16]. Previous studies have reported that although breastfeeding is the primary exposure pathway to PFAS for infants, it is a major form of elimination for mothers. Increased levels of PFAS in breast milk are associated with decreased levels in maternal blood [25,26,29]. Additionally, [30] reported that parous mothers had significantly lower PFOA and PFOS levels in their plasma compared to nulliparous women by 40% each, which confirms that breastfeeding is an elimination route from maternal blood.

Various factors have been linked to PFAS in human milk, including maternal age, socioeconomic status, education level, parity, infant sex, and maternal diet [31]. Moreover, a study conducted in Spain in 2021 showed that women who used more personal care products, including skincare and hair products, deodorant, perfume, and cosmetics, had significantly higher amounts of several PFAS in their breast milk [32].

Infants’ exposure to PFAS has become a public health concern since it has been linked to adverse health outcomes that are more likely to occur due to their low weight, immature immune system, and low detoxification capacity [8,33,34,35,36,37]. Previous studies found that exposure to PFOS and PFOA from many sources during infancy is associated with developmental delays, immature cognitive development, immunosuppression, metabolic syndrome, and even cancer [16,38]. Others have also found a consistent association with liver damage, thyroid diseases, and infertility later in life [37,39]. In Taiwan, [10] reported that exposure to PFAS was significantly associated with asthma in children, while [40] concluded that increased exposure to PFAS in the United States was strongly related to developing attention deficit hyperactivity disorder in children. In addition, it was found that serum PFAS showed a consistent inverse association with the immune response to routine immunizations among children aged 5 to 7 years, increasing their risks of infections [41]. Furthermore, a link between miscarriage risk and PFAs has also been reported [33].

Due to the toxicological concerns over PFOS and PFOA, the US Environmental Protection Agency (EPA) reported in 2006 that these contaminants and their precursors should be eliminated by 2015 [42]. In 2009, they were listed under the United Nations Environment Program’s Stockholm Convention as persistent organic pollutants (POPs) [43]. As a result, global companies have phased out their use during the last two decades, and it has been prohibited to use PFAS in the industry without prior EPA review and approval [44,45].

The European Food Safety Authority (EFSA) identified PFAS as emerging toxicants in the food chain. It established tolerable daily intakes (TDI) of 150 ng/kg BW/day and 1500 ng/kg BW/day for PFOS and PFOA, respectively [46]. However, these values were recently edited. The EFSA issued a new safety threshold of 4.4 ng/kg BW/day due to the health risks of being chronically exposed to these toxic chemicals [47].

Regarding human milk, the EFSA set critical values for these contaminants based on the association found between the concentration of PFAS in the plasma and the amount of antibodies against diphtheria and tetanus in one-year-old infants [48]. The threshold value set for perfluorononanoic acid (PFNA) and PFOA is 60 ng/L, and that for perfluorohexane sulfonate (PFHxS) and PFOS is 73 ng/L, while 133 ng/L is the threshold for the sum of the four PFAS [47,48].

In Lebanon, food safety issues are of public health concern [49,50,51]. Among these are the safety of breast milk and infant formula in terms of toxic metals, mycotoxins, and persistent organic pollutants, which were assessed in several studies [52,53,54,55,56,57]. However, to the best of our knowledge, no study was carried out in Lebanon to determine the level of perfluorooctanoic acid and perfluorooctane sulfonate in breast milk. The aim of our study was to assess the levels of PFOA and PFOS in the breast milk of Lebanese lactating mothers and the maternal factors associated with their presence.

## 2. Methods

### 2.1. Information about the Participants

Socio-demographic characteristics were gathered from participating breastfeeding mothers using a culturally tailored questionnaire. In addition, a semi-quantitative food frequency questionnaire (FFQ) was used to get information regarding eating habits of the lactating mothers for different food groups (egg and milk, cereals and millets, nuts and oil seeds, pulses or legumes, dried fruits, meat, fats and oils, and drinks and beverages). The FFQ was based on guidelines of the WHO [58], and the collected data were expressed as the frequency of food consumption per week. Questionnaires were translated to the Arabic language and then back-translated to English by translators. They were pilot tested on ten subjects who met the eligibility criteria of the participants for comprehension prior to conducting the actual study.

### 2.2. Human Milk Samples

A convenient sample of 57 samples of human milk were obtained from breastfeeding mothers across Lebanon between November 2015 and December 2016. Samples (10–75 mL) were collected in sterile plastic containers, transported in an ice box to the laboratory, where they were stored in the freezer at −20 °C until analysis. Before approaching the participants, Institutional Review Board approval was obtained from the Lebanese American University. Furthermore, a letter was received from the Ministry of Public Health in Lebanon to support the recruitment of eligible participants from primary care centers and hospitals. All participants obtained written and signed informed consent before data collection. The latter was conducted following the World Health Organization protocol [58]. To be eligible, the mother should be without any chronic diseases, have a normal pregnancy course, breastfeed one child only, and be available for milk collection within 3–8 weeks of delivery. Moreover, she should have resided in her current region for at least ten consecutive years [59].

### 2.3. Sample Extraction

As per [60], solid-phase extraction was used, with minor modifications, to extract PFCs in milk. The milk samples were diluted 1:1 in phosphate-buffered saline and centrifuged for 15 min at 100 g to remove any cells. Then, 10 mL of diluted breast milk was mixed with 1 ng each of a mixture of internal standards (including 13C4-PFOS, 13C4-PFOA), and then sonicated for 1 h after the addition of 14 mL of formic acid. Subsequently, Oasis weak anion exchange cartridges (WAX; 6 cc, 150 mg; Waters, Milford, MA, USA) were used to extract samples at a rate of 1 drop/s following the preconditioning of the cartridges, which was conducted by passing 3 mL of 0.1% NH_4_OH in methanol, then by 6 mL of methanol in addition to 6 mL of Milli-Q water, following a rate of 1 drop/s. We ensured the cartridges were preserved from drying during the preconditioning and sample-loading. The cartridges were washed right after passing the sample with 6 mL of Milli-Q water and 6 mL of 25 mM sodium acetate/acetic acid buffer (pH 4). Then, they were cleaned by passing 6 mL of 40% methanol in Milli-Q water. Afterwards, the cartridges were dried for 10 min under vacuum (−70 kPa) to get rid of the residual water before eluting with 6 mL of 0.1% NH_4_OH in methanol at 1 drop/s. The eluate was concentrated under nitrogen to near dryness, then reconstituted to 200 uL using 0.1% NH_4_OH in methanol.

### 2.4. Instrumental Analysis

PFCs in breast milk were identified and quantified using high-performance liquid chromatography (2695 separation module) coupled with a Micromass Quattro micro API triple quadrupole mass spectrometer from (Waters, Milford, MA, USA). Then, 10 μL of the extract was injected onto a 2.1 mm × 100 mm, 3.5 μm Xterra MS C18 column (Waters, Milford, MA, USA) at a flow rate of 0.3 ml/min. The mobile phase was 2 mM ammonium acetate/methanol at a flow rate of 300 μL/min. The gradient begun at 10% methanol, amplified to 99% methanol at 12 min, and was kept for 3 min before switching back to 10% methanol and holding for 3 min for a total run time of 18 min. The API was executed in electrospray negative ionization mode with a source temperature of 100 °C, desolvation temperature of 250 °C, desolvation gas flow of 500 L/h, and cone gas flow of 200 L/h. The target compounds were identified using multiple reaction monitoring. Table 1 presents each analyte’s compound-specific MS/MS parameters and mass transitions.

### 2.5. Quality Assurance and Quality Control

Nine target compounds were spiked to determine matrix-spike recoveries of individual PFCs via the analytical procedure in cow’s milk (3% fat content) and breast milk. PFCs were spiked at 80, 200, and 600 pg/mL in cow’s milk (n) 6 for each spiking level) procured from a local retailer. PFCs were spiked at 200 pg/mL in breast milk (n) 6). The samples were extracted and then analyzed by following the procedure described earlier. Recoveries of PFCs spiked in the cow’s milk were between 86 ± 9% (mean ± standard deviation SD) and 104 ± 3. Recoveries of PFCs spiked in the breast milk were 75 ± 14% to 100 ± 11 for all PFCs, except for LPFBS where the recoveries were 68 ± 35%. Recoveries of the two 13C-labeled internal standards, spiked into all samples before extraction, were between 72 ± 11% and 80 ± 7%. Procedural blanks (n = 6) were made by substituting 10 mL of Milli-Q water for the milk. This was followed by a complete analytical procedure. The average contamination level in procedural blanks was 80 pg/mL for PFOA and <10 pg/mL for the other PFCs. Quantification was conducted using linear regressions (r2 > 0.99 for all analytes) obtained from an eight-point calibration standard, which was prepared in methanol at concentrations between 20 pg/mL and 600 pg/mL. The limit of quantitation (LOQ) was 10 times the standard deviation (SD) of the highest concentration found in the blanks. In comparison, the limit of detection (LOD) was determined as three times the (SD).

### 2.6. Statistical Analysis

STATA V13 was used for the statistical analysis. To summarize the study variables and to detect out-of-range values, descriptive analysis was used. Using frequencies and percentages, categorical variables were described, while means and standard deviations were used in order to represent continuous variables. Shapiro–Wilk was used to assess data normality. Median and Interquartile ranges (IQR) were used to describe the non-parametric variables. Spearman’s rank correlations coefficient (rho) was used to measure the association between the non-parametric variables PFOA, PFOS, age, and BMI. First, the collected human milk samples were divided into two groups for PFOA contamination. Group one included the samples with PFOA levels below the threshold of 60 pg/mL, whereas group two included those with PFOA levels greater than 60 pg/mL. None of the samples had a PFOA level of 60 pg/mL. χ^2^ tests were used to assess the difference in PFOA contamination in terms of consumption of the different food sources such as bread and pasta, cereals, white tubers and roots, potatoes, dry beans, canned beans, fresh vegetables and fruits, dried fruits, meats and chicken, eggs, fish low in mercury and high in mercury, pasteurized milk, sweets, yogurt and labneh, white cheeses, nuts and seeds, spices and condiments, herbal infusions, soft and caffeinated drinks. Second, the human milk samples were divided into two groups for PFOS contamination. Group one included the samples with PFOS levels below the threshold of 73 pg/mL, whereas group two included those with PFOS greater than 73 pg/mL. Considering only 6 participants had PFOS values above the threshold, the analysis was conducted on the non-parametric variable without categorization. Mann Whitney U was conducted to assess the difference in PFOS contamination between the two consumption levels of the previously mentioned food sources.

## 3. Results and Discussion

Demographic characteristics of the participants are shown in Table 2. The mean age ± SD and BMI ± SD of the participants were 28.15 ± 5.01 and 23.09 ± 2.99, respectively. Mount Lebanon (29.6%), Beirut (25.9%), and the South (25.9%) were the main governorates of residence. Most nursing mothers had university degrees (75.9%), 55% were pregnant for the second time, and 28.1% reported complications during pregnancy, mainly hyperemesis gravidarum, gestational diabetes, and iron deficiency anemia. Regarding smoking and smoke exposure, 31.6% reported that they used to smoke before pregnancy, with only 13% continuing to do so during pregnancy. However, the majority were exposed to second-hand smoke (69.8%).

PFOA contamination was detected in 47 (83%) human milk samples. Among the latter, the median PFOA was 147 (IQR 120–274). There was no difference in PFOA contamination in human milk by any socio-demographic variables. Females who consumed bread/pasta and meat/chicken more than twice per week had PFOA levels greater than 60 pg/mL in their breast milk versus those who consumed it less frequently (84.4% vs. 50% *p* = 0.017 and 87% vs. 50% *p* = 0.009, respectively). As for the white tubers and roots, at least once weekly consumption was associated with PFOA contamination (86.4% vs. 50% *p* = 0.010). There was no association between PFOA and other food sources, as indicated in Table 3.

PFOS contamination was detected in six (10.71%) breast milk samples. The median PFOS level was 22 (IQR 13–48.5) among all tested samples. There was no difference in PFOS contamination in human milk across the governorates (*p* = 0.217). Cereal consumption at least twice per week was associated with greater PFOS contamination in breast milk compared to those who consumed it less than twice per week (z = 2.480, *p* = 0.0131).

To the best of our knowledge, this is the first study in Lebanon to assess PFOA and PFOS in human milk and to study the demographic and maternal dietary factors associated with this contamination. Out of the 57 collected human milk samples, PFOA was reported in 83% of the samples ranging between 120 and 247 pg/mL with a median of 147 pg/mL, exceeding the threshold set by the EFSA (60 pg/mL). Concerning PFOS, it was reported in 86% of the samples ranging between 12 and 86 pg/mL, with a median of 27.5 pg/mL, below the threshold set by the EFSA for PFOS in human milk (73 pg/mL).

Compared to the literature, the levels of PFOA in the collected human milk samples in Lebanon were higher than that reported in a neighboring country, Jordan, where the level of contamination ranged between 24–1220 pg/mL with a median of 82.5 pg/mL, compared to a median of 147 pg/mL in Lebanon. However, our samples were less contaminated with PFOS than in Jordan, with a median of 50 pg/mL compared to 27.5 pg/mL in our study [61]. Similarly, the PFOA contamination in our collected human milk was the highest compared to previously conducted studies in the USA, Spain, China, France, and Ireland, with median contamination levels of 13.9 pg/mL, 26 pg/mL, 34.5 pg/mL, 75 pg/mL, and 100 pg/mL, respectively. On the other hand, our reported levels of PFOS were comparable to those reported in Ireland, with a median contamination level of 20 pg/mL. However, our results were the lowest compared to several studies conducted in the USA, China, France, Germany, and Hungary, with median contamination levels of 30.4 pg/mL, 49 pg/mL, 79 pg/mL, 119 pg/mL, and 330 pg/mL, respectively [9,16,62,63,64,65].

Having the highest contamination level in our samples for PFOA and the lowest for PFOS, compared to the literature, can be explained by a recent report in Lebanon that showed legal actions being taken to control the use of PFOS in the industry. At the same time, other PFAS substances, which include PFOA, are still poorly regulated [66]. According to a recent government inventory in 2017, most companies in Lebanon that belong to various industrial sectors, including paper and packaging, textile, carpentry, painting, and plastic products, are not using PFOS-containing substances anymore. This is reflected in our study that has shown low levels of PFOS in the human milk samples. However, PFAS is still mainly present in the imported firefighting foams and is considered a major source of exposure for the Lebanese people; it was estimated that 56 to 167 kg of PFOA or PFOS were released between 2004 and 2014, increasing the exposure of the Lebanese population to these hazardous chemicals [66].

There was no statistical significance between maternal age and human milk contamination with PFOA and PFOS, which is inconsistent with the literature. In Jordan, it was shown that the average contamination levels of PFOA and PFOS were significantly higher in the breast milk of older women [61]. This was also the case in Spain and Korea, where it has been shown that older mothers were consistently found to have higher concentrations of PFAS in their breast milk than younger ones, due to their extended periods of exposure to these contaminants, in addition to the long half-lives of these contaminants in the human plasma [9,67]. This might be because the mean maternal age in our study was 28 years, with only three mothers being older than 35 years old.

Concerning BMI, there are inconsistent findings in the literature. In our study, there was no statistical significance between maternal BMI before pregnancy and the level of contamination. Since the majority of the enrolled nursing mothers in the study had normal BMIs (23.09 ± 2.99 kg/m^2^), this might have affected the results. Our findings are consistent with other studies in France and eastern Slovakia, where no significant association was reported [30,63]. However, in Korea, pre-pregnancy maternal BMI was significantly correlated with the level of PFAS in human milk [67].

Primiparous women have constantly been shown to have higher contamination levels of PFAS in their breast milk than multiparous ones [8,30,67]. According to [9], multiparous women had lower concentrations of PFAS in their breast milk by 58% due to a more significant transfer of these contaminants from maternal plasma to breast milk during the first pregnancy compared to multiple pregnancies. This suggests that firstborns are more at risk of exposure to these toxic contaminants through breast milk [8]. However, our study showed no significant association between PFAS/PFOS contamination and parity. This could be due to the lack of variability in terms of parity among our participants.

In previously conducted studies, women with higher socioeconomic status and greater education levels had increased levels of PFAS in their breast milk [14,30,68,69]. According to [68], women with tertiary education usually belong to a high socioeconomic class, meaning they can afford high-quality products such as waterproof footwear and sportswear, increasing their exposure to PFAS. Interestingly, a recent study showed that high-income countries use more industrial chemicals and have higher exposure to PFAS [33]. However, in Lebanon, there was no significant correlation between maternal education level and the level of contamination. This might be attributed to the lack of variability in the education variable making it difficult to detect a difference.

In addition, the association between levels of PFOA and PFOS in human milk and maternal residence and work environment, in terms of the presence of nearby cultivation activity, gas stations, industrial plants, electrical generators, and living on the main road or beside waste disposal, was assessed. However, no significant association was found. According to the literature, some PFAS are more prevalent in certain occupations than others. For example, PFOS is mostly used in military and firefighting devices. It has been shown that workers exposed to these environments have higher serum levels of PFOS and PFHxS [70]. Occupational workers have been shown to have a higher risk of exposure to these toxic contaminants than the general population [71].

Moreover, several studies showed that the place of residence greatly impacts the levels of PFOA and PFOS in human milk. Higher levels of PFAS were reported in the breast milk of mothers living in urban or semi-urban areas compared to rural ones [40,62,72]. This is consistent with a previously conducted study in China among 12 provinces, which reported a positive correlation between the concentrations of PFAS in human milk and the level of industrialization and economic development [64]. Furthermore, no statistical significance was found between the contamination levels and the governorate where the nursing mothers resided.

There was no statistical significance between PFOA and PFOS contamination and any smoking variables, including smoking before or during pregnancy, random smoking exposure, and household smoking. Similarly, in France, no significant association was found between the levels of PFOA and PFOS in human milk and smoking [63]. Interestingly, a study in eastern Slovakia suggested that smoking during pregnancy was significantly associated with 11–17% lower concentrations of PFAS in breast milk compared to non-smokers. However, self-reported smoking might have influenced the results [30].

Maternal diet is one of the most important factors related to the amount of PFOA and PFOS present in human milk. In this study, those who consumed bread and pasta, meat, and chicken more than twice per week and consumed tubers and roots at least once per week had significantly higher levels of PFOA in their breast milk (*p* = 0.017, *p* = 0.09, and *p* = 0.010, respectively). Regarding PFOS, it was reported to be higher in the milk samples of women who consumed cereals at least twice per week (*p* = 0.0131). However, published studies reported that increased fish and seafood consumption before and during pregnancy was the main dietary factor positively associated with higher contamination of these chemicals in human milk [9,30]. [30] reported that women who consume more than 10 g of fish per day had 20 to 30% higher levels of PFAS in their breast milk. In addition, it was reported that the consumption of butter and margarine was inversely associated with the levels of PFOA and PFOS. On the other hand, a recent study in Spain showed that high levels of PFOS in human milk were associated with increased fried food consumption. The authors suggested that this might be related to the excess use of non-stick cookware or frying with contaminated oil [32].

To conclude, our study is the first in Lebanon to assess the presence of PFOS and PFOA in the breast milk of Lebanese mothers and study the factors associated with their contamination. Additionally, using the ICP-MS technique to assess the levels of these contaminants in the human milk samples is a strength of the study. However, some limitations should be carefully considered when interpreting our results. The main limitation is that our sample is small and might not be representative, affecting the generalizability of our results. Additionally, our study might have a self-reporting bias, where mothers self-reported their dietary intake, smoking status, and alcohol consumption. Future studies must evaluate the circulating/serum PFOS and PFOA levels to be coupled with the levels in human milk.

Our results suggest the need for further investigations with a larger and more representative sample to examine the sources of contamination in Lebanon [73], and assess the infants’ estimated daily intake of these contaminants. Moreover, future studies should also assess the association between the frequency of use of personal care products and the levels of PFOA and PFOS in human milk, since these products are commonly used by women in Lebanon and have shown to be a significant source of exposure for PFAS in many previously conducted studies.

## 4. Conclusions

Human milk is the ideal source of nutrition for infants, essential for their proper growth and development. Milk contamination with PFOA and PFOS poses a serious health threat, hindering the growth of infants, and increasing their risk of chronic diseases later in life. Therefore, infants’ exposure to PFOA and PFOS through human milk is a global health concern. Our study has shown that 86% and 83% of our samples were contaminated with PFOS, and PFOA, respectively. Since human milk is the primary source of infant nutrition, it is essential to ensure its safety by minimizing the mother’s and baby’s exposure to these environmental contaminants, and maximizing the benefits of lactation. Nevertheless, it is important to highlight that exclusive breastfeeding protects nursing infants from exposure to various contaminants, and thus the benefits of breastfeeding compensate for the potential risks. In addition, we should acknowledge that human milk is just one of many routes of exposure of infants to these harmful contaminants. Raising awareness about the importance of choosing healthy and safe food, and avoiding contaminated foods and water, is essential to ensure the health of our infants in the country. Our findings strongly suggest the need for an integrated collaboration between different public health stakeholders to protect the Lebanese population from these toxic chemicals that have lifelong health consequences. It is important to highlight that the benefits of breastfeeding outweigh the reported contamination with PFOS and PFOA in our study.

## Figures and Tables

**Table 1 ijerph-20-00821-t001:** Mass transitions and compound-specific parameters.

Compound	Precursor Ion	Product Ion (*m*/*z*)	Cone (V)	Collision Energy (eV)
**LPFBS**	299	80	40	25
**PFHxA**	313	269	12	8
**PFHpA**	363	319	15	8
**LPHxS**	399	80	45	32
**PFOA**	413	369	15	8
**MPFOA (IS)**	417	372	16	8
**PFNA**	463	419	15	9
**L-PFOS**	499	80	55	45
**MPFOS (IS)**	503	80	55	45
**PFDA**	513	469	15	10
**L-PFDS**	599	80	60	55

**Table 2 ijerph-20-00821-t002:** Demographic characteristics of study participants.

*Demographic Characteristic*	*Mean (SD)*
**Age**	28.15 (5.01)
**BMI (preconception)**	23.09 (2.99)
*Demographic Characteristic*	N (%)
**Governorate**	
North	3 (5.56%)
Bekaa	3 (5.56%)
Nabatiyeh	4 (7.41%)
South	14 (25.93%)
Mount Lebanon	16 (29.63%)
Beirut	14 (25.93%)
**Level of education**	
Illiterate	1 (1.85%)
Primary	2 (3.70%)
Intermediate	2 (3.70%)
Secondary	4 (7.41%)
Technical	4 (7.41%)
University	41 (75.93%)
**Work**	
**Yes**	24 (45.28%)
**Medication Use**	
**Yes**	10 (17.54%) medical conditions reported included anemia, hyperthyroidism, bacteremia, and venous insufficiency
**Supplementation Use**	
Yes	45 (78.95%) mainly multivitamin, folic acid, iron, calcium, and vitamin D
**Complications during pregnancy**	
**Yes**	16 (28.07%)
**Workup in dentures**	
**Yes**	10 (18.52%) mean number of teeth with amalgam 3.9 (SD = 1.57)
**Alcohol Consumption**	
No	54 (98.18%)
**Smoking before pregnancy**	
**Yes**	18 (31.6%)
**Smoking during pregnancy**	
**Yes**	7 (13%)
**Smoking in Household**	
Yes	20 (35.71%)
**Exposed to smoking randomly**	
Yes	37 (69.81%)
**Smoking exposure at work**	
**Yes**	2 (4.08%)

Note. Data are presented as mean ± SD for continuous variables and N (%) for categorical variables.

**Table 3 ijerph-20-00821-t003:** Crosstabulation of food groups and PFOA contamination in Human milk.

Food Group	Overall Sample	PFOA Contamination < 60 pg/mL	PFOA Contamination > 60 pg/mL	Chi-Square Tests of Independence
**Bread and Pasta N (%)** *≤ twice per week* *> twice per week*	12 45	5 (50) 5 (50)	7 (15.6) 38 (84.4)	χ^2^ (1) = 5.6907 *p=* 0.017 ***
**Cereals N (%)** *< twice per week* *≥ twice per week*	14 41	3 (30) 7 (70)	11 (24.4) 34 (75.6)	χ^2^ (1) = 0.1331 *p* = 0.715
**White Tuber and roots N (%)** *< once per week* *At least once per week*	11 42	5 (50) 5 (50)	6 (13.6) 38 (86.4)	χ^2^ (1) = 6.6423 *p =* 0.010 *
**Potatoes N (%)** *At least once per week* *> once per week*	18 37	3 (30) 7 (70)	15 (33.3) 30 (67.7)	χ^2^ (1) = 0.0413 *p = 0.839*
**Dry Beans N (%)** *< once per week* *At least once per week*	25 30	4 (40) 6 (60)	21 (46.7) 24 (53.3)	χ^2^ (1) = 0.1467 *p =* 0.702
**Canned Beans N (%)** *< once per week* *At least once per week*	32 23	4 (40) 6 (60)	28 (62.2) 17 (37.8)	χ^2^ (1) = 1.6606 *p =* 0.198
**Fresh Vegetables Consumption N (%)** *≥ twice per week* *Daily*	25 28	5 (50) 5 (50)	20 (46.5) 23 (53.5)	χ^2^ (1) = 0.0396 *p* = 0.842
**Fresh fruits Consumption N (%)** *≥ twice per week* *Daily*	26 29	4 (40) 6 (60)	22 (48.9) 23 (51.1)	χ^2^ (1) = 0.2594 *p =* 0.611
**Dried fruits Consumption N (%)** *< once per week* *At least once per week*	30 25	3 (30) 7 (70)	27 (60) 18 (40)	χ^2^ (1) = 2.9700 *p =* 0.085
**Pasteurized Milk Consumption N (%)** *< twice per week* *≥ twice per week*	30 25	5 (50) 5 (50)	25 (55.6) 20 (44.4)	χ^2^ (1) = 0.1019 *p =* 0.750
**White Cheeses Consumption N (%)** *≤ twice per week* *Daily*	32 23	4 (40) 6 (60)	28 (62.2) 17 (37.8)	χ^2^ (1) = 1.6606 *p =* 0.198
**Egg N (%)** *once per week* *> once per week*	29 24	3 (33.3) 6 (66.7)	26 (59) 18 (41)	χ^2^ (1) = 2.0007 *p =* 0.157
**Fish Low in Mg N (%)** *< once per week* *At least once per week*	33 22	5 (50) 5 (50)	28 (62.2) 17 (37.8)	χ^2^ (1) = 0.5093 *p =* 0.475
**Fish High in Mg N (%)** *< once per week* *At least once per week*	37 17	5 (50 5 (50)	32 (72.7) 12 (27.3)	χ^2^ (1) = 1.9511 *p =* 0.162
**Meat and Chicken Consumption N (%)** *< twice per week* *≥ twice per week*	11 44	5 (50) 5 (50)	6 (13) 39 (87)	χ^2^ (1) = 6.8750 *p =* 0.009 *
**Spices and condiments N (%)** *< twice per week* *≥ twice per week*	30 18	6 (66.7) 3 (33.3)	24 (61.5) 15 (38.5)	χ^2^ (1) = 0.0821 *p =* 0.775
**Herbal Infusions N (%)** *< twice per week* *≥ twice per week*	26 27	3 (30) 7 (70)	23 (53.5) 20 (46.5)	χ^2^ (1) = 1.7911 *p =* 0.181
**Nuts and Seeds N (%)** *< twice per week* *≥ twice per week*	34 21	4 (40) 6 (60)	30 (66.7) 15 (33.3)	χ^2^ (1) = 2.4650 *p =* 0.116
**Sweets N (%)** *< twice per week* *≥ twice per week*	32 23	6 (60) 4 (40)	26 (57.8) 19 (42.2)	χ^2^ (1) = 0.0166 *p =* 0.897
**Caffeinated drinks N (%)** *≤ twice per week* *Daily*	30 25	6 (60) 4 (40)	24 (53.3) 21 (46.7)	χ^2^ (1) = 0.1467 *p =* 0.702
**Soft Drinks N (%)** *< once per week* *At least once per week*	30 20	5 (55.6) 4 (44.4)	25 (61) 16 (39)	χ^2^ (1) = 0.0903 *p =* 0.764

Note. * *p* < 0.05.

## Data Availability

Available upon request.

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
