# Peer review of "Perfluorooctanoic Acid and Perfluorooctane Sulfonate in Human Milk: First Survey from Lebanon"

_ijerph, 2023, doi:10.3390/ijerph20010821_

Round 1

Reviewer 1 Report

This work investigates levels of PFOA and PFOS in the human milk of Lebanese lactating mothers and the maternal factors associated with their presence. It is interesting to research to be published in the journal. The authors are suggested to specify the locality (city or sector) within Lebanon where the women found came from, considering the limited number of samples.

Author Response

The authors are suggested to specify the locality (city or sector) within Lebanon where the women found came from, considering the limited number of samples.

Thank you for the comment. Location of women within Lebanon was already stated in Table 1.  

Reviewer 2 Report

In the manuscript "Perfluorooctanoic acid and perfluorooctane sulfonate in human milk: first survey from Lebanon" the authors present the first results of the first survey done in Lebanon on the levels of PFOA and PFOS in human milk. The authors find that there was no significant association between PFOA/PFOS with age, BMI,  level of education, place of residence, water, or smoking. However, they found a significant association between  PFOA/PFOS and different food produces and consumables. This an important survey that calls for actions to improve the local environmental and agricultural practices and the regulations and standards for inspecting imported food. The survey is well-planned and the results are presented well.

Author Response

Thank you for the positive feedback.

Reviewer 3 Report

The text is well written and the topic is very interesting.

However, the approach must be corrected to avoid the message that breast milk is a source of risk to the health of the baby and the effect of reducing breastfeeding rates, which are already not very reassuring and the effect would be even more detrimental to the health of infants in the short and long term. For example, artificial milk and its possible contamination is mentioned only once in line 145 without any clarification and without bibliographical references. It is a crucial point because a disinformation campaign is being carried out precisely on this issue. I have not read anywhere in the article that the benefits of breast milk still exceed the limits. The title is also misleading and lends itself to disparaging slogans.

In paragraphs 111 to 121 it should be better specified that these are not effects related to maternal milk but to exposure during infancy. It says but it is not clear.

The passage on smoking in line 373-379 should also be clarified because the message is misleading.

In what period was the study carried out? What kind of study is it? Despite the extensive description, this figure is not evident.

Too bad that the sample is not very representative for the little variability.

The associations with the sources of the two contaminants (food, cosmetics, etc.) should be better summarized. They are described in a confusing manner.

The limits you describe should be indicated separately and not in the discussion.

Many of the journals cited are veterinary medicine and I don't know their IF. Could you point me to some?

Author Response

I have not read anywhere in the article that the benefits of breast milk still exceed the limits. The title is also misleading and lends itself to disparaging slogans.

Thank you for the comment. The following was added to the conclusion: "Nevertheless, it is important to highlight that exclusive breastfeeding protects nursing infants from exposure to various contaminants and thus the benefits of breastfeeding compensate for the potential risks. In addition, we should acknowledge that human milk is just one of many routes of exposure of infants to these harmful contaminants."

In paragraphs 111 to 121 it should be better specified that these are not effects related to maternal milk but to exposure during infancy. It says but it is not clear.

Thank you for the comment. "from different sources" was added to L112.

In what period was the study carried out? What kind of study is it? Despite the extensive description, this figure is not evident.

Thank you for the comment. "between November 2015 and December
2016" was added to L156. In addition, "a convenient sample" was added to L155.

Too bad that the sample is not very representative for the little variability.

Thank you for the comment. We agree on this and it was stated in the limitations L392.

Many of the journals cited are veterinary medicine and I don't know their IF. Could you point me to some?

Thank you for the comment. Two references are from "Veterinary World" journal, which is a Q2 journal, indexed in both Scopus and Web of Science. Its impact factor is 1.98.

Reviewer 4 Report

Journal IJERPH (ISSN 1660-4601)

Manuscript ID ijerph-2074760

Type Article

Title Perfluorooctanoic acid and perfluorooctane sulfonate in human milk: first survey from Lebanon

Authors Hussein F. Hassan , Haneen Bou Ghanem , Joelle Abi Kharma , Mohamad G. Abiad , Jomana Elaridi , Maya Bassil *

Section Health Behavior, Chronic Disease and Health Promotion

Special Issue Ensure Healthy Lives and Promote Wellbeing for All at All Ages

This is an interesting work focused on harmful chemical compounds perfluorooctanoic acid (PFOA) and perfluorooctane sulfonate (PFOS) and their impact on human healthy. The authors have particularly evaluated the levels of PFOA and PFOS in the human milk of Lebanese lactating mothers (n=57) and related these levels in the context of the maternal characteristics/lifestyle factors, such as maternal age, BMI, parity level of education, place of residence, source of water used, and smoking. The median contamination for PFOA exceeded the threshold set by 30 the European Food Safety Authority (EFSA) (60 pg/ml), while that of PFOS was below the 31 threshold (73 pg/ml). The work is interesting and well written. The topic is very important, and I believe it matches well with the journal aims and scope of International Journal of Environmental Research and Public Health. Please see below several comments for improving the work. 

Major comments

1. The International Journal of Environmental Research and Public Health guidelines indicate the separation between results and discussion. However, whether the combining between results and discussion is suitable for publication, it should be an Editor decision

2. One of the main limitations of this work is that it is lacking in important references in the field. Several important papers on the implication of PFOS and PFOAs should be mentioned: DOI: 10.1021/jf304680j, https://doi.org/10.1289/EHP10359, https://doi.org/10.1016/j.scitotenv.2022.154888, https://doi.org/10.1016/j.envint.2012.10.001, https://link.springer.com/article/10.1007/s11783-022-1541-8. If possible, authors should also compare their results with those reported in the literature and discuss their findings in this context  

3. Methods. I suggest switching the 2.1 and 2.2 sections as the study population should be firstly presented I the methods. Moreover, besides the already included inclusion criteria, exclusion criteria should also be mentioned in the methods

4 Please included supporting references in these methodological sections 2.3, 2.4, 2.5, 2.6

5 Have the authors evaluated the circulating/serum PFOS and PFOA levels in the study cohort? this might be an interesting information that might be coupled with the already reported milk concentrations 

6 among limitations, the small sample size should be clearly mentioned

Minor observations

Lines 47-49 PFASs are still under use in several countries, with serious consequences (doi: 10.1039/d0em00291g), please include this notion. 

Lines 51-57 the effect of PFOA ahs been evaluated in an animal model-based study which evaluated the expression of several target genes in carps exposed to environmental doses of PFOA (DOI: 10.1002/etc.4029). Thus underlining that even in conditions of low concentrations, the exposure to compounds can negatively impact the expression of genes in animals an possibly humans. This important notion should be, at least briefly, included and the work quoted. 

Line 86 If possible, more recent WHO guidelines should be cited. Those from 2011 seems to be quite old 

Lines 109-121 a miscarriage risk and PFAs has also been reported https://link.springer.com/article/10.1007/s11783-022-1541-8

Please included supporting references in this methodological section

Line 217 it should be standard deviation (SD) the first time this noun is mentioned in the text. Plese do the same in line 267 for ”interquartile range (IQR)”

Line 253 “mean +/- SD age and BMI were…”

Line 299 citations?

Author Response

The International Journal of Environmental Research and Public Health guidelines indicate the separation between results and discussion. However, whether the combining between results and discussion is suitable for publication, it should be an Editor decision.

Thank you for the comment. We leave it to the Editor to advise if we should combine Results and Discussion in one section.

One of the main limitations of this work is that it is lacking in important references in the field. Several important papers on the implication of PFOS and PFOAs should be mentioned: DOI: 10.1021/jf304680j, https://doi.org/10.1289/EHP10359, https://doi.org/10.1016/j.scitotenv.2022.154888, https://doi.org/10.1016/j.envint.2012.10.001, https://link.springer.com/article/10.1007/s11783-022-1541-8

Thank you for the comment. Actually, doi: 10.1016/j.envint.2012.10.001 is already cited. We cited the other 4 articles.

Methods. I suggest switching the 2.1 and 2.2 sections as the study population should be firstly presented I the methods. Moreover, besides the already included inclusion criteria, exclusion criteria should also be mentioned in the methods

Thank you for the comment. 2.1 and 2.2 got switched. Recruiting participants was according to WHO protocol (2007), which does not state exclusion criteria.

Have the authors evaluated the circulating/serum PFOS and PFOA levels in the study cohort? this might be an interesting information that might be coupled with the already reported milk concentrations 

Thank you for the comment. No, we did not. We added this to the recommendations for future research (L407-408).

Among limitations, the small sample size should be clearly mentioned.

Thank you for the comment. "small" was added to L405.

Lines 47-49 PFASs are still under use in several countries, with serious consequences (doi: 10.1039/d0em00291g), please include this notion.

Thank you for the comment. This notion got included. Also, Gluge et al. (2020) got cited.

Lines 51-57 the effect of PFOA ahs been evaluated in an animal model-based study which evaluated the expression of several target genes in carps exposed to environmental doses of PFOA (DOI: 10.1002/etc.4029). Thus underlining that even in conditions of low concentrations, the exposure to compounds can negatively impact the expression of genes in animals an possibly humans. This important notion should be, at least briefly, included and the work quoted. 

Thank you for the comment. This notion was added. Rotondo et al. (2018) got cited as well.

Line 86 If possible, more recent WHO guidelines should be cited. Those from 2011 seems to be quite old

Thank you for the comment. In fact, WHO guidelines regarding exclusive breastfeeding did not change since 2011. 

Lines 109-121 a miscarriage risk and PFAs has also been reported https://link.springer.com/article/10.1007/s11783-022-1541-8

Thank you for the comment. This notion was added and Fiedler et al. (2022) got cited.

Line 217 it should be standard deviation (SD) the first time this noun is mentioned in the text. Plese do the same in line 267 for ”interquartile range (IQR)”

Thank you for the comment. Done for the SD. As for the IQR, it was mentioned already as "interquartile range" the first time. 

Line 253 “mean +/- SD age and BMI were…”

Thank you for the comment. Done.

Line 299 citations?

Thank you for the comment. "(Innovation Development Association, 2019)" was added.